# MELK expression correlates with tumor mitotic activity but is not required for cancer growth

Christopher J Giuliano[1†], Ann Lin[1†], Joan C Smith[2], Ann C Palladino[1], Jason M Sheltzer[1]*

[1]Cold Spring Harbor Laboratory, Cold Spring Harbor, United States; [2]Google, Inc., New York, United States

**Abstract** The Maternal Embryonic Leucine Zipper Kinase (MELK) has been identified as a promising therapeutic target in multiple cancer types. MELK over-expression is associated with aggressive disease, and MELK has been implicated in numerous cancer-related processes, including chemotherapy resistance, stem cell renewal, and tumor growth. Previously, we established that triple-negative breast cancer cell lines harboring CRISPR/Cas9-induced null mutations in MELK proliferate at wild-type levels in vitro (*Lin et al., 2017*). Here, we generate several additional knockout clones of MELK and demonstrate that across cancer types, cells lacking MELK exhibit wild-type growth in vitro, under environmental stress, in the presence of cytotoxic chemotherapies, and in vivo. By combining our MELK-knockout clones with a recently described, highly specific MELK inhibitor, we further demonstrate that the acute inhibition of MELK results in no specific anti-proliferative phenotype. Analysis of gene expression data from cohorts of cancer patients identifies MELK expression as a correlate of tumor mitotic activity, explaining its association with poor clinical prognosis. In total, our results demonstrate the power of CRISPR/Cas9-based genetic approaches to investigate cancer drug targets, and call into question the rationale for treating patients with anti-MELK monotherapies.
DOI: https://doi.org/10.7554/eLife.32838.001

*For correspondence:
sheltzer@cshl.edu

†These authors contributed equally to this work

## Introduction

Cancer cells require the expression of certain genes, called 'addictions' or 'genetic dependencies', that encode proteins necessary for tumor growth. Silencing the expression of these genes or blocking the activity of the proteins that they encode can trigger cell death and durable tumor regression (*Luo et al., 2009*). Identifying and characterizing cancer dependencies is therefore a key goal of pre-clinical cancer research.

The Maternal Embryonic Leucine Zipper Kinase (MELK) has been implicated as a cancer dependency and putative drug target in multiple cancer types, including melanoma, colorectal cancer, and triple-negative breast cancer (*Chung et al., 2012*; *Speers et al., 2016*; *Wang et al., 2014*; *Ganguly et al., 2014a*; *Ganguly et al., 2014b*; *Janostiak et al., 2017*). MELK is over-expressed in these cancers, and high expression of MELK is associated with poor patient prognosis (*Wang et al., 2014*; *Pickard et al., 2009*; *Phillips et al., 2006*; *Ryu et al., 2007*; *Gray et al., 2005*). Moreover, knockdown of MELK using RNA interference (RNAi) has been reported to block cancer cell proliferation and trigger cell cycle arrest or mitotic catastrophe (*Speers et al., 2016*; *Wang et al., 2014*; *Gray et al., 2005*; *Kuner et al., 2013*; *Lin et al., 2007*; *Beke et al., 2015*). On the basis of these pre-clinical results, several companies have developed small-molecule MELK inhibitors, and one MELK inhibitor (OTS167) is currently being tested in multiple clinical trials (*ClinicalTrials.gov, 2017*).

In contrast to these results, we recently reported that triple-negative breast cancer cells harboring CRISPR/Cas9-induced loss-of-function mutations in MELK proliferate at wild-type levels in vitro (*Lin et al., 2017*). Additionally, we demonstrated that the MELK inhibitor OTS167 remained effective against MELK-mutant cells, suggesting that OTS167 kills cells through an off-target mechanism. These results have been replicated by an independent group, who further demonstrated that the shRNA vectors commonly used to study MELK also kill cells in a MELK-independent manner (*Huang et al., 2017*). The off-target effects of both the small-molecule MELK inhibitor and the MELK-targeting shRNAs provide a potential explanation for certain previous results obtained studying this reported drug target.

Despite the conflicting in vitro data, MELK expression remains one of the strongest predictors of patient mortality in diverse cancer types (*Smith and Sheltzer, 2017*). Additionally, MELK has been implicated in several other cancer-related processes, including cancer stem cell maintenance, chemotherapy resistance, anchorage-independent growth, and reactive oxygen species (ROS)-signaling (*Wang et al., 2014*; *Ganguly et al., 2014a*; *Beke et al., 2015*; *Kim et al., 2015*; *Seong et al., 2016*; *Choi et al., 2011*; *Gu et al., 2013*; *Hebbard et al., 2010*). These processes may not be challenged by the routine in vitro growth assays that have been performed in MELK-knockout (MELK-KO) cells to date. Several previous studies were also conducted using distinct RNAi constructs and small-molecule inhibitors, raising the possibility that they reflect true functions of this kinase. Moreover, the over-expression of MELK has been reported to transform cells, suggesting that in addition to MELK's putative role as a cancer dependency, it may also function as a driver oncogene (*Wang et al., 2014*).

While RNA interference is susceptible to off-target interactions that confound experimental interpretation (*Jackson et al., 2003*, *2006*), CRISPR/Cas9 mutagenesis is also prone to several important limitations. In particular, clonal cell lines harboring Cas9-induced modifications must be expanded from a single cell with a mutation of interest to several million cells. This intense pressure may select for secondary mutations that blunt any anti-proliferative consequences of the original mutation (*Iwasa et al., 2006*). In a therapeutic context, the immediate inhibition of a particular target achieved with a small-molecule drug may induce a more severe phenotype than observed in a CRISPR-modified cell line subjected to evolutionary pressure over the course of days or weeks.

To investigate the role of MELK in cancer-related processes beyond cell proliferation, and to assess the therapeutic potential of immediate MELK inhibition, we performed assays combining CRISPR-knockout cell lines with a recently described, highly specific MELK inhibitor (*Huang et al., 2017*). In a variety of in vitro and in vivo challenges, we found that cells lacking MELK behave indistinguishably from wild-type cells. Moreover, through a close analysis of gene expression data, we report that MELK levels strongly correlate with mitotic activity in human tumors, suggesting that MELK expression may function as an indirect proxy for rapid cell division. In total, these results cast doubt on the possibility that MELK-specific inhibition will serve as a useful monotherapy in cancer.

## Results

### MELK over-expression fails to transform immortalized cell lines

The over-expression of driver oncogenes allows immortalized but non-transformed cell lines to form colonies when grown in soft agar, a phenotype that is tightly linked with in vivo tumorigenicity (*Shin et al., 1975*; *Colburn et al., 1978*; *Cifone and Fidler, 1980*). It has previously been reported that the over-expression of MELK was sufficient to induce anchorage-independent growth in several cell lines, including Rat1 fibroblasts expressing dominant-negative p53 (p53dd) and the human mammary epithelial cell line MCF10A (*Wang et al., 2014*). We attempted to replicate these results using Rat1-p53dd and MCF10A cells, as well as the immortalized 3T3 mouse fibroblast cell line. To accomplish this, we stably transduced each cell line with a retroviral vector encoding either the mouse or the human MELK protein. Western blot analysis confirmed that full-length mouse or human MELK was over-expressed in all six cell lines that we generated relative to the level of MELK in vector-transduced control cell lines (*Figure 1A,D*). Consistent with previous reports (*Gray et al., 2005*; *Badouel et al., 2010*), we found that both ectopic and endogenous MELK were stabilized and

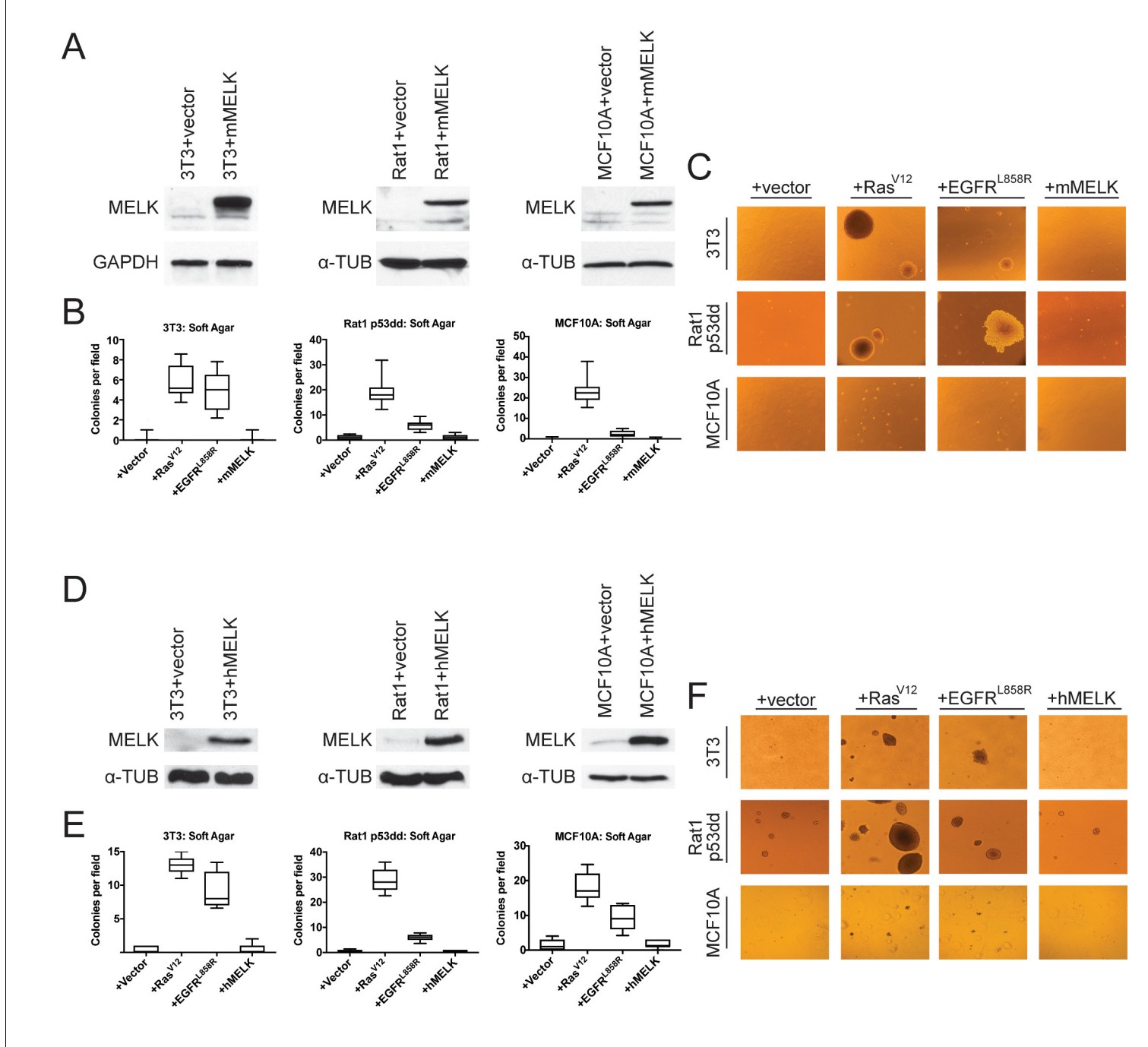

**Figure 1.** MELK over-expression fails to confer anchorage-independent growth. (A) Western blot analysis of mouse MELK over-expression in 3T3, Rat1-p53dd, and MCF10a cell lines. (B) Quantification of colony formation of control and mouse MELK over-expressing cell lines in soft agar. For each assay, colonies were counted in at least 15 fields under a 10x objective. Boxes represent the 25th, 50th, and 75th percentiles of colonies per field, while the whiskers represent the 10th and 90th percentiles. (C) Representative images of the indicated cell lines grown in soft agar. (D) Western blot analysis of human MELK over-expression in 3T3, Rat1-p53dd, and MCF10a cell lines. (E) Quantification of colony formation of control and human MELK over-expressing cell lines in soft agar. For each assay, colonies were counted in at least 15 fields under a 10x objective. Boxes represent the 25th, 50th, and 75th percentiles of colonies per field, while the whiskers represent the 10th and 90th percentiles. (F) Representative images of the indicated cell lines grown in soft agar.

DOI: https://doi.org/10.7554/eLife.32838.002

The following figure supplement is available for figure 1:

**Figure supplement 1.** Ectopically-expressed MELK exhibits a phospho-shift in mitosis.

*Figure 1 continued on next page*

*Figure 1 continued*

DOI: https://doi.org/10.7554/eLife.32838.003

phosphorylated in mitotic cells, suggesting that the transgenic MELK was functional (*Figure 1—figure supplement 1*).

As positive controls in the anchorage-independent growth assay, we transduced each cell line with an allele of H-Ras known to function as a strong driver oncogene (H-Ras$^{G12V}$)(*Tabin et al., 1982*), and with an allele of EGFR that weakly transforms cells (EGFR$^{L858R}$) (*Greulich et al., 2005*). We then assessed whether cell lines over-expressing each gene would proliferate when suspended in soft agar. As expected, cells that had been transduced with an empty vector exhibited minimal anchorage-independent growth, while cells transduced with H-Ras$^{G12V}$ or EGFR$^{L858R}$ formed numerous colonies (*Figure 1B–C and E–F*). However, in multiple independent experiments, we failed to detect an increase in anchorage-independent growth in any of the six cell lines over-expressing either mouse or human MELK. We conclude that, under the conditions tested, the over-expression of MELK fails to transform cells.

## MELK is dispensable for growth in vitro and in vivo

We previously showed that MELK was not required for cell division in two triple-negative breast cancer cell lines, Cal51 and MDA-MB-231 (*Lin et al., 2017*). However, MELK has been reported to support growth in several other cancer types, including colorectal cancer and melanoma (*Ganguly et al., 2014b*; *Janostiak et al., 2017*; *Gray et al., 2005*; *Choi et al., 2011*). To examine the role of MELK in other cancer types, we used CRISPR/Cas9 to generate multiple MELK-knockout (MELK-KO) clones in A375, a melanoma cell line, and DLD1, a colorectal cancer cell line (*Figure 2—figure supplement 1*). As controls, we also derived clones of A375 and DLD1 harboring guide RNAs that targeted the non-essential Rosa26 locus. MELK mutagenesis was verified by sequencing the sites targeted by the gRNA, and loss of the MELK protein was verified by western blotting with two antibodies that recognize distinct epitopes (*Figure 2—figure supplement 1A–B*). MELK-KO melanoma and colorectal cancer clones grew at wild-type levels in vitro, demonstrating that MELK is dispensable for proliferation in these cancer types as well (*Figure 2—figure supplement 1C*). We previously reported that OTS167, a putative MELK inhibitor in clinical trials, kills breast cancer cells in a MELK-independent manner (*Lin et al., 2017*). Consistent with these observations, MELK-KO and Rosa26 clones were equally sensitive to OTS167, verifying that this drug kills cells via an off-target effect across cancer types (*Figure 2—figure supplement 1D*).

Many genes that are non-essential for cell division in vitro may still play crucial roles in cancer by supporting other processes, including stem-cell renewal, resistance to anoikis, and angiogenesis (*Rotem et al., 2015*; *Zhong et al., 2011*; *Miller et al., 2017*; *Cidado et al., 2016*). We therefore subjected our MELK-KO and Rosa26 clones to various in vitro and in vivo assays to assess whether MELK loss impairs any cancer-related phenotypes. Although all MELK-KO cells grow well when seeded at high density in proliferation assays (*Lin et al., 2017*; *Huang et al., 2017*), plating cells at low density can challenge a cell's colony-forming ability and replicative lifespan (*Franken et al., 2006*). To test whether MELK loss confers a defect in colony growth, MELK-KO and Rosa26 A375, Cal51, DLD1, and MDA-MB-231 clones were serially diluted and allowed to grow at varying cell densities. Crystal violet staining of these plates revealed that the loss of MELK failed to impair colony growth relative to the control Rosa26 cell lines (*Figure 2A*). In fact, one MELK-KO clone (MDA-MB-231 c1) grew consistently better than either control clone in this assay. Significant variation in proliferative capacity has previously been described among independent clones of the MDA-MB-231 cell line (*Khan et al., 2017*) and may arise due to heterogeneity in the parental population or secondary mutations acquired during cell line derivation.

To further investigate the impact of MELK loss on anoikis and tumorigenicity, MELK-KO and Rosa26 cells were plated in soft agar. However, we observed no significant difference in anchorage-independent growth between MELK-KO cells and Rosa26 cells in every cell line tested (*Figure 2B–C*). MELK has previously been implicated in the maintenance of breast cancer stem cells (*Ganguly et al., 2014a*; *Gu et al., 2013*; *Kig et al., 2013*). We therefore tested whether loss of MELK impairs mammosphere formation, a phenotype tightly linked with breast cancer stem cell

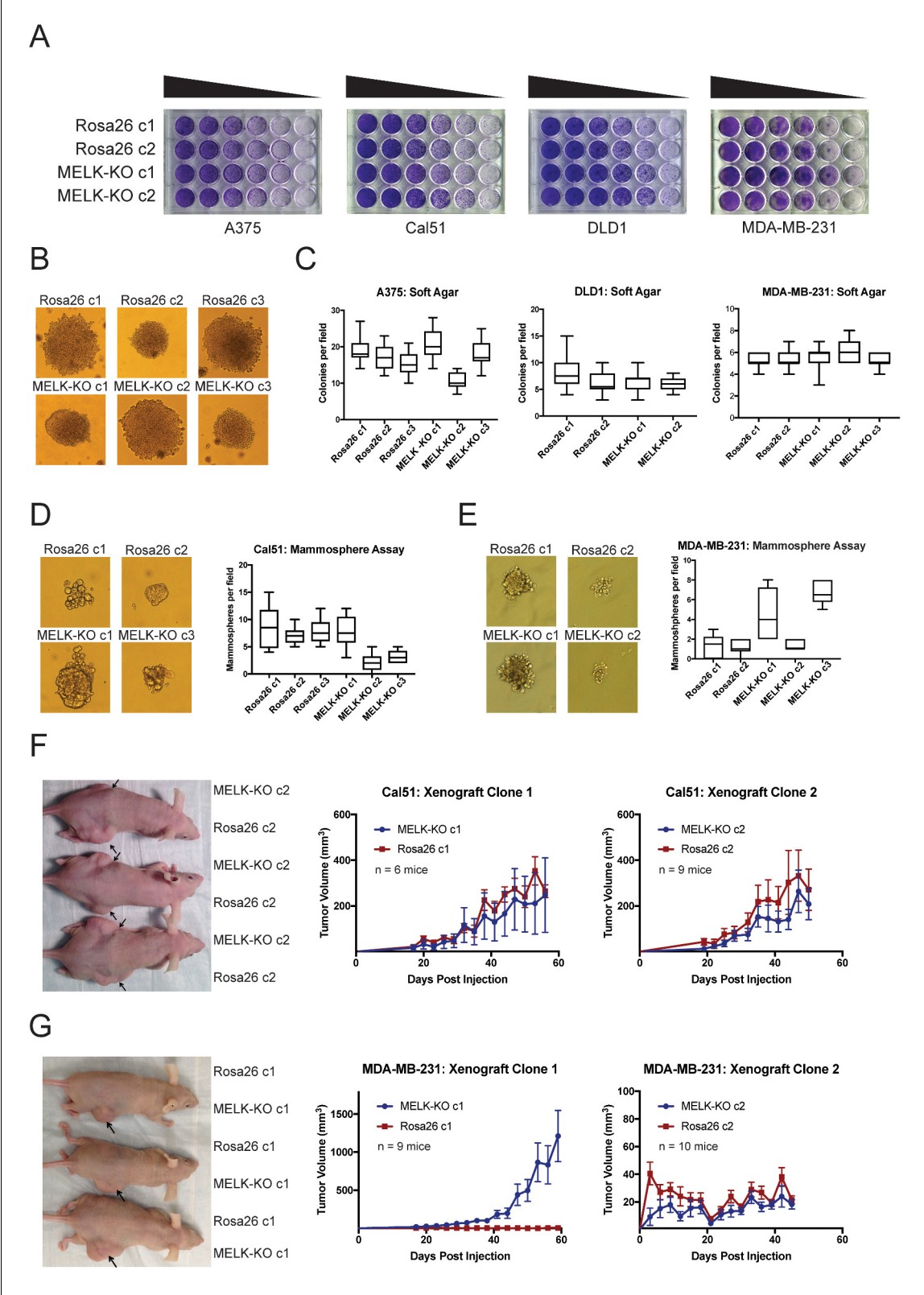

**Figure 2.** MELK is dispensable for growth in vitro and in vivo. (A) Crystal violet staining of serial dilution plates of control and MELK knockout clones from A375 (melanoma), Cal51 (breast cancer), DLD1 (colorectal cancer), and MDA-MB-231 (breast cancer) cell lines. (B) Representative images of colonies of A375 control and MELK knockout clones grown in soft agar. (C) Quantification of colony formation in A375, DLD1, and MDA-MB-231 MELK-KO and control clones. For each assay, colonies were counted in at least 15 fields under a 10x objective. Boxes represent the 25th, 50th, and 75th

*Figure 2 continued on next page*

*Figure 2 continued*

percentiles of colonies per field, while the whiskers represent the 10th and 90th percentiles. (**D–E**) Representative images and quantification of mammosphere growth in Cal51 or MDA-MB-231 MELK-KO and control clones. For each assay, mammospheres were counted in at least six fields under a 10x objective. Boxes represent the 25th, 50th, and 75th percentiles of colonies per field, while the whiskers represent the 10th and 90th percentiles. (**F–G**) Representative images and quantification of xenograft growth in nude mice. Cal51 and MDA-MB-231 MELK-KO and control clones were injected subcutaneously into nude mice, and then tumor growth was measured every 3 days. Arrows indicate the location of the tumor. Error bars in the volume measurements indicate the standard error.

DOI: https://doi.org/10.7554/eLife.32838.004

The following figure supplements are available for figure 2:

**Figure supplement 1.** Generation and characterization of MELK-KO clones in A375 and DLD1.

DOI: https://doi.org/10.7554/eLife.32838.005

**Figure supplement 2.** MELK is dispensable for tumor formation in heterogeneous MDA-MB-231 populations.

DOI: https://doi.org/10.7554/eLife.32838.006

---

activity (*Grimshaw et al., 2008*). While we observed inter-clonal variability in mammosphere growth, all MELK-KO clones were capable of forming mammospheres, and two MELK-KO clones exhibited consistently greater mammosphere formation than their wild-type controls (*Figure 2D–E*). We conclude that MELK is dispensable for growth as a mammosphere.

Finally, we sought to determine whether MELK was required for tumor formation in vivo. To address this, we performed flank injections into nude mice with multiple clones of Cal51 and MDA-MB-231 MELK-KO and Rosa26 cells. Across all clones tested, 23 of 34 injections with MELK-KO cells and 24 of 34 injections with Rosa26 cells resulted in detectable tumor formation, and no significant growth defect was observed in any MELK-KO clone (*Figure 2F–G*). Consistent with our previous assays, we observed superior growth in the MDA-MB-231 MELK-KO c1 clone, while MDA-MB-231 MELK-KO c2 and Rosa26 c2 grew slowly but at equivalent rates (*Figure 2G*). The slow growth of MDA-MB-231 Rosa26 c1, MELK-KO c2, and Rosa26 c2 was unexpected, as MDA-MB-231 is reported to grow aggressively in vivo (*Price et al., 1990*). We hypothesized that the poor growth of many of our MDA-MB-231 clones could reflect the fact that cells in the parental population exhibit different abilities to form tumors, and by chance, we isolated several clones with limited tumor-initiating capacity (*Fillmore and Kuperwasser, 2008*; *Charafe-Jauffret et al., 2009*). To further explore the role of MELK in tumorigenesis, we transduced the parental MDA-MB-231 population with guide RNAs targeting either Rosa26 or MELK, and then selected the gRNA-expressing populations without single-cell cloning. Western blot analysis verified that transduction with the MELK-targeting gRNAs ablated MELK protein expression (*Figure 2—figure supplement 2A*). After injection into nude mice, these cell populations grew more rapidly than any clonal cell line, and the MELK-depleted cells exhibited equivalent or superior tumor growth compared to the control populations (*Figure 2—figure supplement 2B*). In total, these results demonstrate that MELK is dispensable for the proliferation of cancer cells in vitro and in vivo.

## MELK is not required for the phosphorylation or expression of previously reported targets

MELK has been reported to support cancer cell proliferation by phosphorylating various proteins involved in splicing, translation, metabolism, and cell cycle progression (*Seong et al., 2016*; *Wang et al., 2016*; *Vulsteke et al., 2004*; *Seong et al., 2002*; *Joshi et al., 2013*). In particular, a recent publication reported that MELK phosphorylates eukaryotic translation initiation factor 4B (eIF4B), and this phosphorylation event promotes cell survival by increasing translation of the anti-apoptotic protein MCL1 (*Wang et al., 2016*). However, western blot analysis revealed normal levels of eIF4B phosphorylation in MELK-knockout A375, Cal51 and MDA-MB-231 cells (*Figure 3—figure supplement 1A–C*). Additionally, MELK-KO cell lines continued to express MCL1, the putative downstream target of eIF4B (*Figure 3—figure supplement 1D–E*). We conclude that MELK is not required for eIF4B phosphorylation or MCL1 translation.

## MELK is dispensable for cell growth under exogenous stress

Developing tumors must survive in hypoxic and nutrient-poor conditions (*Vaupel et al., 1989*), and MELK has been implicated in glucose signaling and in the detection of ROS (*Seong et al., 2016*;

*Jung et al., 2008*). We therefore considered the possibility that MELK expression is necessary to support growth under metabolic or environmental stress. To generate ROS stress, we cultured cells in varying concentrations of $H_2O_2$, but we observed no difference in sensitivity between MELK-KO and control Rosa26 clones (*Figure 3A*). MELK has been suggested to contribute to ROS signaling by phosphorylating ASK1 (Jung et al., 2008); however, this protein remained phosphorylated at normal levels in cells lacking MELK (*Figure 3—figure supplement 1F*). MELK-KO clones also exhibited wild-type levels of growth when cultured under hypoxic, serum-deprived, or glucose-limited conditions (*Figure 3B–D*). We conclude that MELK is dispensable for proliferation under common metabolic stresses.

As high MELK expression is associated with poor patient prognosis, we wondered if MELK could promote resistance to cytotoxic chemotherapies. Indeed, it has been previously reported that knocking down or inhibiting MELK sensitizes cells to DNA damage (*Beke et al., 2015*; *Kim et al., 2015*; *Choi et al., 2011*; *Kig et al., 2013*). To test whether MELK has a role in chemotherapy resistance, we performed drug sensitivity assays in our MELK-KO and Rosa26 clones using a variety of DNA-damaging or anti-mitotic agents. However, we found that the loss of MELK failed to sensitizes cells to five common chemotherapies (*Figure 3E–I*). In total, these results demonstrate that MELK is dispensable for cell survival under metabolic and cytotoxic stress.

## Acute inhibition of MELK fails to block proliferation

Deriving CRISPR-knockout clones from single cells selects for a population of cells that are capable of surviving clonal expansion. Since our MELK-KO cell lines were generated from single cells, we considered the possibility that MELK plays an important role supporting proliferation, but the clones we generated had evolved to tolerate the loss of MELK. To assess this possibility, we performed an 'epistasis' experiment combining our MELK-knockout clones with a recently described, highly-specific small molecule MELK inhibitor, HTH-01–091 (*Huang et al., 2017*). We reasoned that treating Rosa26 clones with HTH-01–091 would reveal the consequences of the acute loss of MELK. However, if such phenotype(s) were also present in MELK-KO cell lines treated with HTH-01–091, then the phenotype(s) could be attributed to an off-target effect of the drug.

We first sought to identify a concentration at which HTH-01–091 inhibited MELK in our cells of interest. However, as described in this manuscript, we lacked a verified MELK substrate whose phosphorylation status could be monitored to confirm MELK inhibition. Nonetheless, it has been reported that a by-product of MELK inhibition is the degradation of MELK protein (*Beke et al., 2015*; *Huang et al., 2017*). Therefore, to determine an effective concentration of HTH-01–091, we monitored the level of MELK protein after drug treatment by western blot. We found that 1 µM HTH-01–091 triggered near-complete MELK degradation in Cal51, DLD1, and MDA-MB-231 cells (*Figure 4A*). The loss of MELK was not an indirect effect of cell cycle arrest, as these cells maintained high levels of the mitotic marker cyclin B. This concentration was therefore used in subsequent assays.

To test whether the acute inhibition of MELK affected clonogenicity or anchorage-independent growth, we grew MELK-KO and Rosa26 cells on plastic or in soft agar in the presence of DMSO or 1 µM HTH-01–091. Neither MELK-KO nor Rosa26 clones were affected by HTH-01–091 treatment, verifying that MELK is dispensable for colony formation and anchorage-independent growth in these cells (*Figure 4B–C*). Indeed, a drug sensitivity assay revealed that HTH-01–091 exhibited significant anti-proliferative effects only at concentrations above ~5 µM (*Figure 4D*). This toxicity is likely a consequence of an off-target effect, as these drug concentrations were found to affect Rosa26 and MELK-KO cells equivalently.

We next sought to test whether acute MELK inhibition sensitized cells to chemotherapy. To accomplish this, we treated MELK-KO and Rosa26 clones with various chemotherapy drugs in the presence of HTH-01–091. Consistent with our previous results, HTH-01–091 treatment failed to sensitize the Rosa26 clones to 5-florouracil or paclitaxel treatment (*Figure 4E*). Finally, we assessed the effect of HTH-01–091 treatment on eIF4B phosphorylation and MCL1 expression, and found that MELK inhibition failed to affect either target (*Figure 3—figure supplement 1G–H*). In total, these results demonstrate that the acute loss of MELK results in no significant defect in cell viability, proliferation, or drug resistance and suggest that our knockout clones have not acquired mutations that tolerize cells to the loss of MELK.

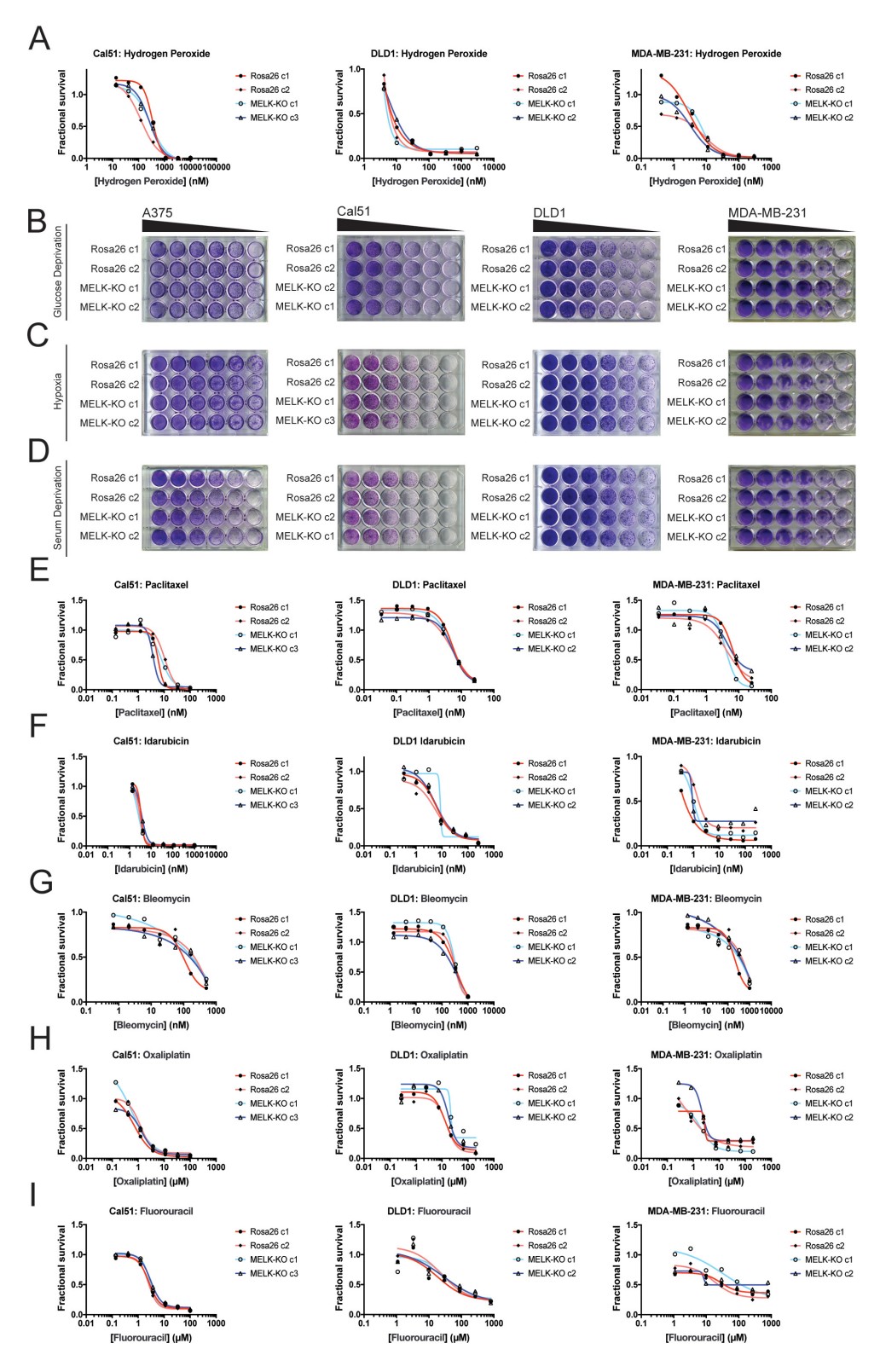

**Figure 3.** MELK is not required for growth under stress. (**A**) Dose-response curves of Cal51, DLD1, and MDA-MB-231 Rosa26 and MELK-KO clones grown in the presence of $H_2O_2$. (**B–D**) Crystal violet staining of A375, Cal51, DLD1, and MDA-MB-231 Rosa26 and MELK-KO clones grown as serial dilutions under the indicated stressful culture condition. (**E–I**) Dose-response curves of Cal51, DLD1, and MDA-MB-231 Rosa26 and MELK-KO clones grown in the presence of the indicated chemotherapy drug.

*Figure 3 continued on next page*

*Figure 3 continued*

DOI: https://doi.org/10.7554/eLife.32838.007

The following figure supplement is available for figure 3:

**Figure supplement 1.** MELK is not required for the phosphorylation or expression of previously-reported targets.

DOI: https://doi.org/10.7554/eLife.32838.008

## The association between MELK expression and cancer lethality is due to its correlation with mitotic activity

Our in vitro and in vivo experiments failed to reveal any cancer-related phenotypes affected by either the deletion or over-expression of MELK. Yet, MELK is up-regulated in many cancer types (*Gray et al., 2005*), and high levels of MELK expression have been reported to confer a dismal clinical prognosis (*Wang et al., 2014*; *Pickard et al., 2009*; *Phillips et al., 2006*; *Ryu et al., 2007*). If MELK plays no overt role in cancer biology, then why would MELK expression be linked with death from cancer? We note that MELK expression is cell cycle-regulated, peaking in mitosis (*Wang et al., 2014*; *Badouel et al., 2010*), and gene signatures that capture mitotic activity have been found to be prognostic in multiple cancer types (*Smith and Sheltzer, 2017*; *Venet et al., 2011*; *Gentles et al., 2015*). We therefore considered the possibility that, rather than functioning as an oncogene or a cancer dependency, MELK expression could report cell division within a tumor. To assess the link between MELK expression and cell division, we analyzed gene expression data from different sets of cells and tissues. In normal human tissue, *MELK* transcript expression was the lowest in non-proliferative organs, including the heart and skeletal muscle, while *MELK* expression was the highest in organs with on-going mitotic activity, including the bone marrow and testes (*Figure 5A*). Indeed, across 32 tissue types, *MELK* levels were highly correlated with the expression of the proliferation marker *MKI67* (R = 0.93). Similarly, in human fibroblasts cultured until senescence, *MELK* expression decreased up to 11-fold between proliferating and arrested populations, while stimulating lymphocytes to divide increased *MELK* expression 20-fold (*Figure 5B–C*). These data suggest that *MELK* levels reflect mitotic activity in diverse cell types.

To examine the link between MELK and cell division in cancer, we compared the levels of *MELK* expression with five well-characterized proliferation markers: *MKI67, PCNA, CCNB1, MCM2,* and *TOP2A* (*Whitfield et al., 2006*). In cohorts of patients with tumor types in which MELK levels have previously been associated with advanced disease, *MELK* expression was significantly correlated with each of the proliferation genes (median correlation = 0.82; *Figure 5D*). We then sought to determine whether the correlation between *MELK* expression and proliferation could explain the prognostic significance of MELK. To test this, we collected 15 breast cancer microarray datasets from patients with known clinical outcomes. For each patient cohort, we calculated Z scores from univariate Cox proportional hazards models, which assess the significance of a putative prognostic variable. A Z score greater than 1.96 indicates that increasing expression of a gene is associated with dismal prognosis at a p<0.05 threshold (*Gentles et al., 2015*). We found that *MELK* expression was significantly linked with poor outcome in 14 of 15 datasets (*Supplementary file 3*). Similarly, a proliferation meta-gene derived by averaging the normalized expression of *MKI67, PCNA, CCNB1, MCM2,* and *TOP2A* was significantly associated with poor outcome in 13 of 15 datasets (*Supplementary file 3*). Strikingly, we found that the Z scores generated by our univariate proliferation models were significantly correlated with the Z scores generated by the univariate *MELK* models: in patient cohorts in which proliferation was highly-prognostic, *MELK* expression was also highly-prognostic, and vice-versa (*Figure 5E*; R = 0.78, p<0.001). These analyses suggested that *MELK* expression and cell proliferation capture very similar clinical information. To test whether *MELK* expression remained prognostic when controlling for mitotic activity, we generated bivariate Cox models that included both *MELK* expression and the proliferation meta-gene (*Supplementary file 4*). In the bivariate models, *MELK* was significantly associated with patient outcome in only 2 of 15 datasets, demonstrating that considering tumor cell proliferation ablated *MELK*'s clinical utility (*Figure 5F*). Thus, when tumors are stratified according to their proliferation level, *MELK* expression is no longer prognostic (*Figure 5G*). In total, these results suggest that the observed pattern of *MELK* expression in cancer can be explained by the fact that *MELK* is up-regulated in mitotic cells.

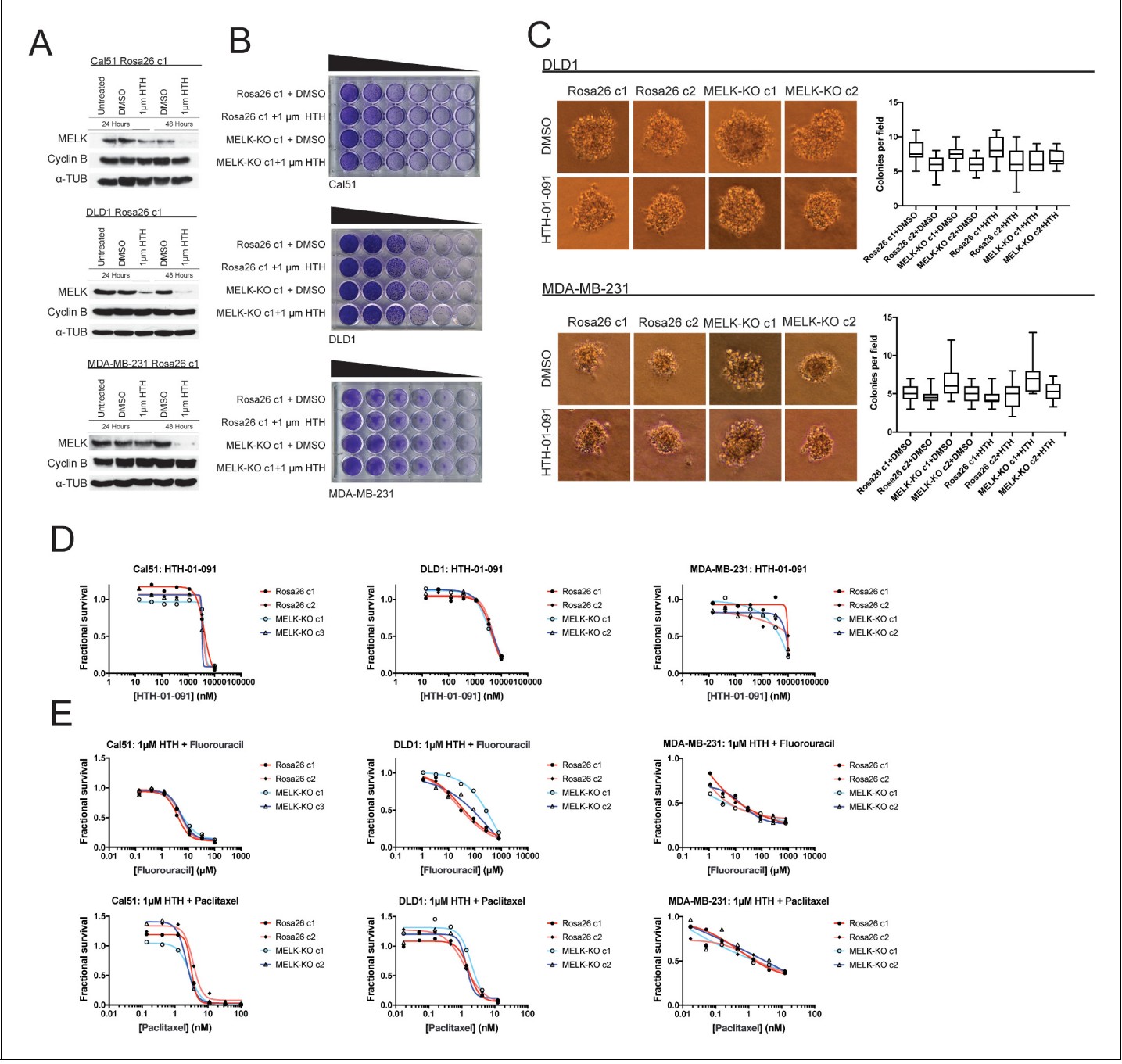

**Figure 4.** Acute inhibition of MELK fails to block growth. (**A**) Western blot analysis of MELK expression levels during treatment with 1 μM HTH-01–091 in the Cal51, DLD1, and MDA-MB-231 Rosa26 clonal cell lines. (**B**) Crystal violet staining of Cal51, DLD1, and MDA-MB-231 Rosa26 and MELK-KO cell lines grown as serial dilutions in the presence of DMSO or 1 μM HTH-01–091. (**C**) Quantification and representative images of colony formation of DLD1 and MDA-MB-231 Rosa26 and MELK-KO clones in soft agar in the presence of DMSO or 1 μM HTH-01–091. (**D**) Dose-response curves of Cal51, DLD1, and MDA-MB-231 Rosa26 and MELK-KO clones grown in the presence of HTH-01–091. (**E**) Dose-response curves of Cal51, DLD1, and MDA-MB-231 Rosa26 and MELK-KO clones grown in the presence of 1 μm HTH-01–091 and the indicated chemotherapy drug.
DOI: https://doi.org/10.7554/eLife.32838.009

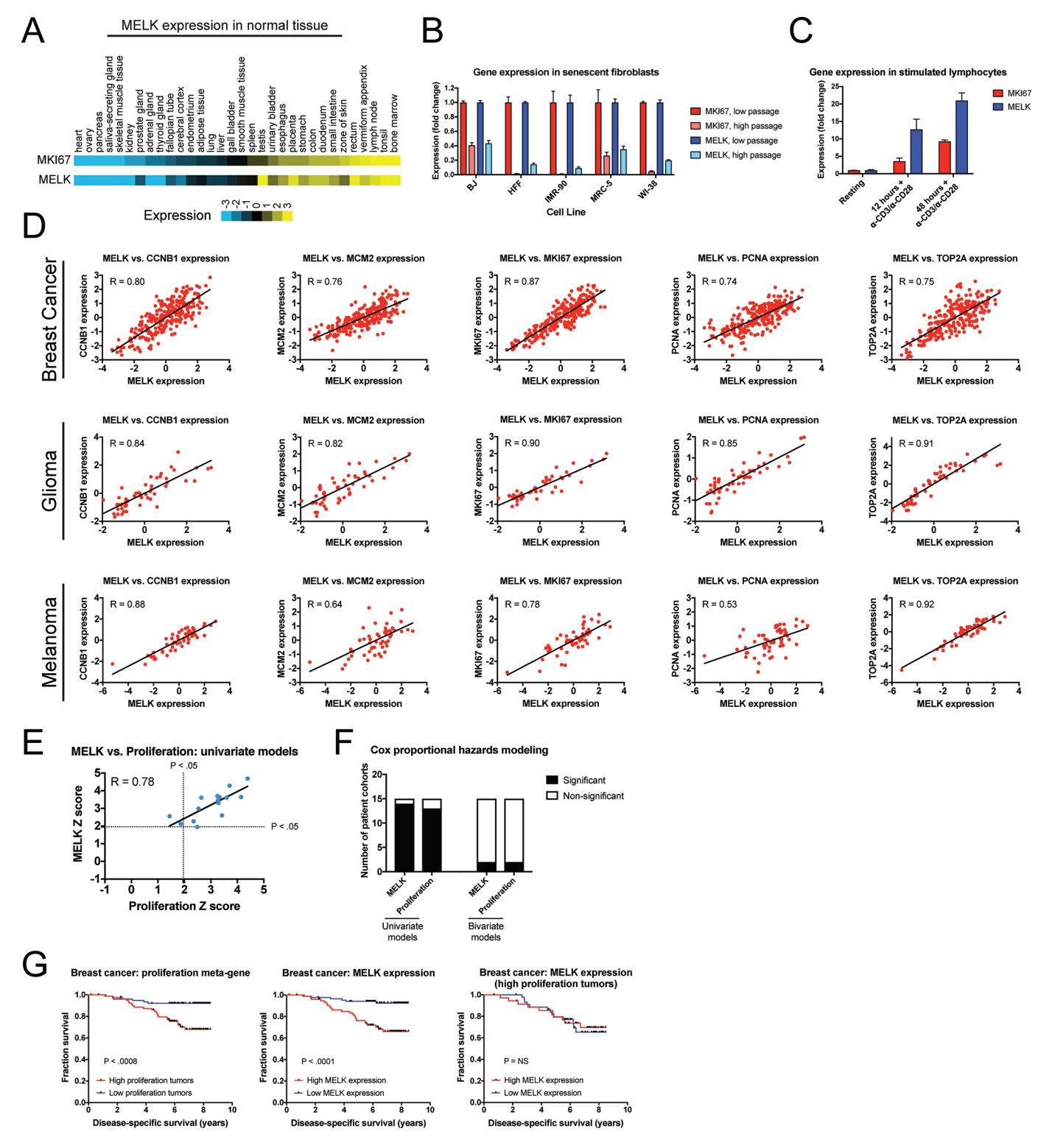

**Figure 5.** *MELK* expression correlates with proliferation markers in vitro, in normal tissue, and in cancer. (**A**) A heatmap of the expression of either *MKI67* or *MELK* in normal tissue sorted according to *MKI67* expression (*Uhlén et al., 2015*). (**B**) The expression level of either *MKI67* or *MELK* is displayed in five different primary human fibroblast lines at either low passage (proliferating) or high passage (senescence) (*Marthandan et al., 2015*). (**C**) The expression level of either *MKI67* or *MELK* is displayed in CD4 + lymphocytes resting or after stimulation with α-CD3/α-CD28 beads (*Abbas et al., 2005*). (**D**) The expression level of *MELK* is plotted against the expression of five common proliferation markers in cohorts of patients

*Figure 5 continued on next page*

*Figure 5 continued*

with breast cancer, glioma, or melanoma (*Sabatier et al., 2011*; *Turcan et al., 2012*; *Jönsson et al., 2010*). Black lines represent linear regressions plotted against the data. (E) Univariate Cox proportional hazards models were calculated for the 15 breast cancer cohorts listed in *Supplementary file 3*. For each cohort, the expression of either *MELK* or the average expression of *CCNB1, MCM2, MKI67, PCNA,* and *TOP2A* in each tumor was regressed against patient outcome. Dotted lines represent Z scores of 1.96, corresponding to a p-value of 0.05. The black line represents a linear regressions plotted against the data. (F) Bar graphs depict the number of cohorts in which *MELK* and a proliferation meta-gene are significantly associated with poor outcome in either univariate or bivariate models. The full results are presented in *Supplementary file 3* and *Supplementary file 4*. (G) Kaplan-Meier curves displaying disease-specific survival in one breast cancer cohort (*Pawitan et al., 2005*). Patients were split into two populations based on the average expression of either *MELK* or the five-gene proliferation meta-gene.

DOI: https://doi.org/10.7554/eLife.32838.010

## Discussion

Pre-clinical cancer research efforts apply different genetic and chemical tools (RNA interference, CRISPR, and small-molecule inhibitors) to a variety of artificial assays (in vitro proliferation, xenograft growth, etc.) in order to discover targets that will have clinical efficacy when inhibited in human patients. No single assay perfectly mimics the behavior of a tumor in a human cancer patient, and no single chemical or genetic tool exhibits absolute specificity. Nonetheless, we believe that by using multiple orthogonal approaches and assays, we can gain insight into the role that certain genes play in human malignancies. In this current manuscript, we report that combining CRISPR and a small-molecule inhibitor in a variety of assays failed to reveal a role for MELK in several cancer-related processes. These results suggest that anti-MELK monotherapies are unlikely to be effective cancer treatments.

Our previous work demonstrated that MELK is dispensable for the proliferation of triple-negative breast cancer cells in vitro (*Lin et al., 2017*). Nonetheless, multiple cellular functions are dispensable for in vitro proliferation but required for tumor progression, including stem cell renewal, oxygen sensing, and chemotherapy resistance (*Rotem et al., 2015*; *Zhong et al., 2011*; *Miller et al., 2017*; *Cidado et al., 2016*). Although MELK has been implicated in each of these processes, our results demonstrate that MELK-knockout cancer cell lines grow at wild-type levels in a variety of assays designed to test these pathways. We speculate that, as has previously been reported for one MELK inhibitor and one set of MELK-targeting shRNA's (*Huang et al., 2017*), several previous studies of MELK function may have been compromised by off-target activity of the constructs and inhibitors that were used.

To assess MELK function in cancer, we used CRISPR/Cas9 to generate mutations in MELK in four different cancer cell lines. Of note, we observed variability between clones from the same parental cell line, and between clonal lines and the parental cell populations. These observations underscore the importance of assessing multiple independent clones and cell populations in CRISPR experiments. Furthermore, to rule out the possibility that our MELK-KO clones had evolved to tolerate the loss of MELK, we performed 'epistasis' experiments by treating these knockout clones with a MELK inhibitor. We reasoned that specific consequences of MELK inhibition would be detectable upon drug treatment in MELK-WT but not MELK-KO clones, while non-specific consequences of drug treatment would affect both genotypes equally. In all experiments conducted thus far, no phenotypes have been observed only in MELK-WT cells after HTH-01–091 treatment, further verifying that MELK is dispensable for cancer cell growth. We suggest that these 'epistasis' experiments can be widely applied to assess the on-target consequences of acutely inhibiting potential cancer drug targets.

Initial interest in blocking MELK function in cancer stemmed from the discovery that it was over-expressed across cancer types (*Gray et al., 2005*). Further research revealed that patients whose tumors expressed the highest levels of MELK had the worst clinical outcomes (*Wang et al., 2014*; *Pickard et al., 2009*; *Phillips et al., 2006*; *Ryu et al., 2007*; *Kuner et al., 2013*). As we observed no role for MELK as either an oncogene or a cancer dependency, we sought to instead investigate whether its cell cycle-dependent expression pattern could explain its prognostic value. Consistent with this hypothesis, we discovered that *MELK* transcript expression closely mirrors tumor mitotic activity. As actively-growing cancers appear to up-regulate thousands of genes involved in cell cycle progression when compared to quiescent normal tissue (*Whitfield et al., 2006*), this observation

may explain why MELK is commonly over-expressed in different malignancies. Moreover, rapid cell division in tumors is indicative of tissue de-differentiation and aggressive disease; therefore, most cell cycle-regulated genes are also associated with poor clinical outcome (*Smith and Sheltzer, 2017*; *Venet et al., 2011*; *Gentles et al., 2015*). We further demonstrated that controlling for cell proliferation ablates the prognostic significance of MELK expression, suggesting that this link may explain its connection with outcome in cancer.

While many cell cycle genes are indeed suitable cancer drug targets (e.g. CDK4 and CDK6), other genes may be up-regulated during normal cell division but dispensable for this process (*Venet et al., 2011*). We suggest that MELK is an example of the latter. We believe that, if MELK does play a role in cancer, it may be detectable only in very limited circumstances, and likely in vivo. As MELK-knockout mice display no observable deficiencies, MELK's function may be redundant with other kinases (*Wang et al., 2014*). Future synthetic lethal screening and additional in vivo assays may clarify what role, if any, MELK plays in cancer biology.

# Materials and methods

## Key resources table

| Reagent type (species) or resource | Designation | Source or reference | Identifiers | Additional information |
|---|---|---|---|---|
| Cell line (human) | A375 | ATCC | RRID:CVCL_0132 | |
| Cell line (human) | Cal51 | Dr. David Solomon and Dr. Todd Waldman | RRID:CVCL_1110 | |
| Cell line (human) | DLD1 | ATCC | RRID:CVCL_0248 | |
| Cell line (human) | MDA-MB-231 | ATCC | RRID:CVCL_0062 | |
| Cell line (rat) | Rat1 | CSHL Cell Line Repository | RRID:CVCL_0512 | |
| Cell line (human) | MCF10A | Dr. Camila Dos Santos | RRID:CVCL_0598 | |
| Cell line (mouse) | 3T3 | CSHL Cell Line Repository | RRID:CVCL_0594 | |
| Cell line (human) | All MELK-knockout and Rosa26 control clonal cell lines (A375, Cal51, DLD1, MDA-MB-231) | This paper and Lin et al., (eLife 2017) | | These cell lines were derived in this paper. Available from Dr. Jason Sheltzer. |
| Transfected construct | LRG 2.1 vector | Dr. Christopher Vakoc and Dr. Junwei Shi | | |
| Transfected construct (human and mouse) | MELK over expression plasmid | Vectorbuilder, Cyagen Corporation | | |
| Antibody | Anti-MELK N terminal Antibody | Abcam | ab108529 | 1:3000 in 5% Milk TBST |
| Antibody | Anti-MELK C terminal Antibody | Cell Signal | 2274S | 1:10000 in 10% BSA TBST |
| Antibody | Anti-ASK1 | Abcam | ab45178 | 1:1000 in 5% Milk TBST |
| Antibody | Anti-eIF4B | Cell Signal | 3592 | 1:2000 in 5% Milk TBST |
| Antibody | Anti-cyclin B | Abcam | ab32053 | 1:10000 in 5% Milk TBST |
| Antibody | Anti-Phospho-ASK1 | Cell Signal | 3765 | 1:1000 in 5% BSA TBST |
| Antibody | Anti-Phospho-eIF4B | Cell Signal | 5399 | 1:1000 in 5% BSA TBST |
| Antibody | Anti-MCL1 | Cell Signal | 5453 | 1:2000 in 5% Milk TBST |
| Antibody | Anti-Alpha-Tubulin | Sigma-Aldrich | T6199 | 1:20000 in 5% Milk TBST |
| Antibody | Anti-GAPDH | Santa Cruz Biotechnology | sc-365062 | 1:20000 in 5% Milk TBST |
| Antibody | Anti-Rabbit | Abcam | ab6721 | 1:50000 to 1:20000 in 5% Milk TBST |
| Antibody | Anti-Mouse | Bio-Rad | 1706516 | 1:50000 in 5% Milk TBST |

*Continued on next page*

*Continued*

| Reagent type (species) or resource | Designation | Source or reference | Identifiers | Additional information |
|---|---|---|---|---|
| Chemical compound, drug | HTH-01–091 | Dr. Nathanael Gray DOI: 10.7554 | | |
| Chemical compound, drug | OTSSP167 | MedChem Express | HY-15512A | |
| Software, algorithm | Survival Analysis | This paper | | https://github.com/joan-smith/survival-analysis-scripts |

## Cell lines and culture conditions

The identity of each human cell line was verified by STR profiling (University of Arizona Genetics Core). A375 (RRID:CVCL_0132), Cal51 (RRID:CVCL_1110), DLD1 (RRID:CVCL_0248), MDA-MB-231 (RRID:CVCL_0062) cell lines were grown in DMEM supplemented with 10% FBS, 2 mM glutamine, and 100 U/mL penicillin and streptomycin. Rat1 (RRID:CVCL_0512) cells were grown in DMEM supplemented with 5% FBS, 2 mM glutamine, and 100 U/mL penicillin and streptomycin. 3T3 (RRID:CVCL_0594) cells were grown in DMEM supplemented with 10% bovine calf serum (BCS), 2 mM glutamine, and 100 U/mL penicillin and streptomycin. MCF10A (RRID:CVCL_0598) cells were grown in Mammary Epithelial Cell Growth Medium (Lonza, Switzerland; Cat. No. CC-3150) supplemented with 5% horse serum, 100 ng/mL of cholera toxin (Sigma-Aldrich, St. Louis, MO; Cat. No. C8052) and 100 U/mL penicillin and streptomycin. All cell lines were grown in a humidified environment at 37°C and 5% $CO_2$.

## Retroviral plasmid over-expression

Mouse and human MELK cDNA was cloned into an MMLV vector and verified by sequencing (Vectorbuilder, Cyagen Corporation). Positive and negative control plasmids were acquired from Addgene: pBabe-Puro (Addgene; Cat. No. 1764), EGFR[L858R] (Addgene; Cat. No. 11012), and Ras[G12V] (Addgene; Cat. No. 1768). Retrovirus was generated by transfecting plasmids into Plat-A cells (Cell BioLabs, San Diego, CA; Cat. No. RV-102) using the calcium-phosphate method (*Smale, 2010*). Virus was harvested 48–72 hr post transfection, filtered through a 0.45 µm syringe and applied to cells with 4 µg/mL polybrene. After 24 hr, the media was changed and cells were allowed to recover in fresh media for 2 days. Subsequently, the cells were split and the appropriate antibiotic was added to select for transduced cells.

## Soft agar assays

To assay anchorage-independent growth, all cell lines except MCF10A were suspended at a cell count of 10,000 cells in a 0.35% Difco Agar Noble (VWR Scientific, USA; Cat. No. 90000–772) solution in a six-well plate. MCF10A cell lines were suspended at a cell count of 20,000 cells. The mixture was plated over a 0.5% Difco Agar Noble solution. Plates were allowed to solidify at room temperature for 1 hr and then placed in a 37°C incubator overnight. 1 mL of normal growth media was added to each well the next day and every 3 days after (*Borowicz et al., 2014*). After 14 days, colony formation was quantified under 20x magnification.

## Stress assays

To study the role of MELK in surviving stressful culture conditions, 30,000 MELK-KO and Rosa26 cells from DLD1 and MDA-MB-231 cell lines and 10,000 MELK-KO and Rosa26 cells from A375 and Cal51 cell lines were plated in the first column of a 24-well plate (Corning, USA; Cat. No. 3526) and then five three-fold dilution were performed across the plate. For the low-glucose conditions, low-glucose DMEM (Thermo Fisher Scientific, Waltham, MA; Cat. No. 11885076) supplemented with 10% FBS, 2 mM glutamine, and 100 U/mL penicillin and streptomycin was added to the cells. For the low serum conditions, Cal51, DLD1, and MDA-MB-231 cell lines were cultured in DMEM supplemented with 1% FBS, 2 mM glutamine, and 100 U/mL penicillin and streptomycin, while the A375 cell line was cultured in 5% FBS, 2 mM glutamine, and 100 U/mL penicillin and streptomycin. For the hypoxic conditions, cells were cultured in normal media and then placed in a hypoxic incubator set at 37°C with

2% oxygen. After 10–14 days, cells were fixed with 100% methanol and stained with 0.5% crystal violet dissolved in 25% methanol.

## Mammosphere formation assay

Mammosphere formation media was prepared using DMEM/F12 (Lonza; Cat. No. CC-3151) supplemented with 2 mM L-glutamine, 100 U/mL penicillin and streptomycin, 20 ng/mL recombinant human epidermal growth factor (Sigma Aldrich; Cat. No. E9644), 10 ng/mL recombinant human basic fibroblast growth factor (R and D Systems; 233-FB-025) and 1x B27 supplement (Invitrogen, Waltham, MA; Cat. No. 17504–044) (*Lombardo et al., 2015*). Cells were plated at a density of 20,000 cells or 30,000 cells per well for the MDA-MB-231 and Cal51 cell lines respectively in a 6-well low attachment plate (Corning; Cat. No. CLS3814) with 3 mL of media. Fresh media was added to the wells every 3 days over the course of the assay. Mammospheres were measured 4 weeks post plating for the MDA-MB-231 cell lines and 2 weeks post plating for Cal51 cell lines.

## Drug sensitivity assays

To quantify a cell line's sensitivity to a particular drug, 10,000 A375, DLD1, or MDA-MB-231 cells or 5000 Cal51 cells were plated in 100 µL of media in an $8 \times 3$ matrix on a flat-bottomed 96-well plate (Corning; Cat. No. 3596). Cells were allowed 24 hr to attach, then fresh media was added to each well. The highest concentration of a drug was added onto the first row of cells and then six three-fold serial dilutions were performed. Cells were grown in the presence of the drug for 72 hr then trypsinized and counted using a MacsQuant Analyzer 10 (Milltenyi Biotec, Germany). Replicate wells were averaged and then normalized to the cell count in the untreated wells. Normalized values were plotted in Prism 7 (Graphpad, San Diego, California) and fit to a curve using a four-parameter inhibition vs. concentration model. HTH-01–091 was a kind gift of Hubert Huang and Nathanael Gray (Dana-Farber Institute). OTSSP167 was obtained from MedChem Express (Monmouth Junction, NJ; Cat. No. HY-15512A). Idarubicin, Oxaliplatin, Paciltacxel and Bleomycin were obtained from Selleck Chemicals (Houston, TX; Cat. No. S1228, S1224, S1150, and S1214). Fluorouracil was obtained from Sigma Aldrich (Cat. No. F6627-1G).

## Western blot analysis

Whole cell lysates were harvested using RIPA buffer (25 mM Tris, pH 7.4, 150 mM NaCl, 1% Triton X 100, 0.5% sodium deoxycholate, 0.1% sodium dodecyl sulfate, protease inhibitor cocktail, and phosphatase inhibitor cocktail). Protein concentration was quantified using the RC DC Protein Assay (Bio-Rad, Hercules, CA; Cat. No. 500–0119) or the Pierce BCA Protein Assay Kit (Thermo Fisher Scientific; Cat. No. 23225). Equal amounts of lysate were denatured and loaded onto an 8% SDS-PAGE gel. The protein was transferred onto a polyvinylidene difluoride membrane using the Trans-Blot Turbo Transfer System (Bio-Rad). Westerns with phospho-antibodies were blocked in 5% BSA, westerns with Anti-MELK C-Terminal (Cell Signal, Danvers, MA; Cat. No. 2274S) were blocked in 10% BSA, and all other antibodies were blocked with 5% milk. The following antibodies and dilutions were used: Anti-MELK N-Terminal (Abcam, Cambridge, MA; Cat. No. ab108529) at a dilution of 1:3000, Anti-MELK C-Terminal (Cell Signal; Cat. No. 2274S) at a dilution of 1:10000, Anti-elF4B (Cell Signal; Cat. No. 3592) at a dilution of 1:2000, Anti-ASK1 (Abcam, Cat. No. ab45178) at a dilution of 1:1000, Anti-cyclin B (Abcam; Cat. No. ab32053) at a dilution of 1:10000, Anti-Phospho-ASK1 (Cell Signal; 3765) at a dilution of 1:1000, Anti-Phospho-elF4B (Cell Signal; Cat. No. 5399) at a dilution of 1:1000, and Anti-MCL1 (Cell Signal; Cat. No. 5453) at a dilution of 1:2000. Blots were incubated with the primary antibody overnight at 4°C. Anti-alpha tubulin (Sigma-Aldrich; Cat. No. T6199) at a dilution of 1:20,000 or Anti-GAPDH (Santa Cruz Biotechnology, Santa Cruz, CA; Cat. No. sc-365062) at a dilution of 1:20,000 were used as loading controls. Membranes were washed at room temperature for an hour before they were incubated in secondary antibodies [Anti-Rabbit (Abcam; Cat. No. ab6721) at 1:50,000 for Anti-MELK and at 1:20,000 for all other antibodies or anti-mouse (Bio-Rad; Cat. No. 1706516) at 1:50,000 for Anti-tubulin and Anti-GAPDH)] for an hour.

## Xenograft growth assays

Nude mice were obtained from The Jackson Laboratory (Bar Harbor, ME; Cat. No. 002019). To perform the xenograft injections, breast cancer cells were harvested and resuspended at a

concentration of $10^8$ cells/mL in 1X cold PBS. The cell suspension was then mixed 1:1 with growth factor reduced-matrigel (Corning; Cat. No. 47743–720). Each mouse was injected subcutaneously in the left and right flanks with 100 μL of the cell suspension, containing $5 \times 10^6$ cells. Tumors were monitored by visual inspection routinely after injection. Once a tumor was visible, mice were measured every 3 days by caliper in duplicate. Tumor volume was calculated using the formula V = ½ (longer axis)(shorter axis)$^2$. All mouse protocols were approved by the CSHL Institutional Animal Care and Use Committee.

## CRISPR plasmid construction and virus generation

Guide RNAs were previously described in Lin et al. (*Lin et al., 2017*). In short, oligonucleotides were cloned into the LRG 2.1 vector [a gift from Junwei Shi (University of Pennsylvania) and Chris Vakoc (Cold Spring Harbor Laboratory)] using a BsmBI digestion (*Shalem et al., 2014*). To produce virus, HEK293T cells were transfected using the calcium-phosphate method (*Smale, 2010*). Supernatant was harvested 48 to 72 hr post-transfection, filtered through a 0.45-μm syringe, and then applied to cells with 4 μg/mL polybrene (*Smale, 2010*). Guide RNAs used to disrupt MELK are listed in *Supplementary file 1*.

## Analysis of CRISPR-mediated mutagenesis

Single cells isolated via fluorescence-activated cell sorting (FACS) were grown into clonal populations. Genomic DNA was extracted from these populations with the QIAmp DNA Mini kit (Qiagen Germantown, MD; Cat. No. 51304). The cut site regions targeted by the guide RNAs were amplified using the primers listed in *Supplementary file 2*. PCR products were then sequenced with the forward and reverse primer at the Cold Spring Harbor Laboratory sequencing facility to yield the 'Cut-site PCR' sequences shown in *Figure 2—figure supplement 1*. To analyze individual alleles, PCR products were ligated into the pCR4-TOPO TA vector from the TOPO TA Cloning Kit (Thermo Fisher Scientific; Cat. No. 450030). Ligated plasmids were transformed into One Shot Stbl3*E. coli* (Thermo Fisher Scientific: Cat. No. C737303). Plasmids from 8 to 20 colonies were extracted using QIAprep Spin Miniprep Kit (Qiagen; Cat. No. 27104) and sequenced with the forward and reverse primer at the Cold Spring Harbor Laboratory sequencing facility.

## Analysis of published gene expression data

Data from normal human tissues were acquired from (*Uhlén et al., 2015*). Data from senescent fibroblasts were acquired from (*Marthandan et al., 2015*). Data from stimulated lymphocytes were acquired from (*Abbas et al., 2005*). Cancer patient cohorts and probeset definitions were downloaded from the Gene Expression Omnibus as described in *Supplementary file 3* (*Edgar et al., 2002*). Data were cleaned and processed using python's pandas library to exclude missing values and to associate clinical outcomes with expression data. To generate the proliferation meta-gene, the expression of five proliferation-related genes (*MKI67, PCNA, CCNB1, TOP2A,* and *MCM2*) were collapsed by averaging. Cox proportional hazard models were constructed using the survival library in R and the coxph function as described in (*Smith and Sheltzer, 2017*). The Cox proportional hazard models for both univariate and bivariate analyses were run from a python script that used rpy2 to run R code from python. Source code is available on github (https://github.com/joan-smith/survival-analysis-scripts; a copy is archived at https://github.com/elifesciences-publications/survival-analysis-scripts)(*Smith, 2016*).

## Acknowledgements

We thank Hubert Huang and Nathanael Gray (Dana-Farber Institute) for providing HTH-01–091. This work was performed with assistance from CSHL Shared Resources, including the CSHL Flow Cytometry Shared Resource, which are supported by the Cancer Center Support Grant 5P30CA045508. Research in the Sheltzer Lab is supported by an NIH Early Independence Award (1DP5OD021385), a Breast Cancer Alliance Young Investigator Award, and a CSHL-Northwell Translational Cancer Research Grant.

## Additional information

### Competing interests

Joan C Smith: Joan C. Smith is affiliated with Google Inc. The author has no financial interests to declare. The other authors declare that no competing interests exist.

### Funding

| Funder | Grant reference number | Author |
| --- | --- | --- |
| NIH Office of the Director | 1DP5OD021385 | Jason M Sheltzer |

The funders had no role in study design, data collection and interpretation, or the decision to submit the work for publication.

### Author contributions

Christopher J Giuliano, Ann Lin, Conceptualization, Investigation, Methodology, Writing—original draft, Writing—review and editing; Joan C Smith, Software, Investigation; Ann C Palladino, Validation, Investigation, Methodology; Jason M Sheltzer, Conceptualization, Data curation, Formal analysis, Supervision, Funding acquisition, Investigation, Methodology, Writing—original draft, Writing—review and editing

### Author ORCIDs

Christopher J Giuliano (iD) https://orcid.org/0000-0002-0586-6095
Jason M Sheltzer (iD) http://orcid.org/0000-0003-1381-1323

### Ethics

Animal experimentation: All mouse protocols were approved by the CSHL Institutional Animal Care and Use Committee (project number 831105-4).

### Decision letter and Author response

Decision letter https://doi.org/10.7554/eLife.32838.056
Author response https://doi.org/10.7554/eLife.32838.057

## Additional files

### Supplementary files

• Supplementary file 1. MELK gRNA sequences. Cancer cells lines used in this study and the guide RNAs that they express are displayed.
DOI: https://doi.org/10.7554/eLife.32838.011

• Supplementary file 2. PCR primers to amplify MELK gRNA cut sites. The sequences of all PCR primers used in this study are displayed.
DOI: https://doi.org/10.7554/eLife.32838.012

• Supplementary file 3. Univariate Cox proportional hazards models of breast cancer outcome and the expression of a proliferation meta-gene or of *MELK*. Cox proportional hazards survival analysis was performed on 15 breast cancer microarray datasets correlating the length of patient survival with either MELK expression or with a proliferation meta-gene comprising *MKI67, PCNA, CCNB1, MCM2,* and *TOP2A*.
DOI: https://doi.org/10.7554/eLife.32838.013

• Supplementary file 4. Bivariate Cox proportional hazards models of breast cancer outcome and the expression of a proliferation meta-gene and of *MELK*. Cox proportional hazards survival analysis was performed on 15 breast cancer microarray datasets correlating the length of patient survival with both MELK expression and with a proliferation meta-gene comprising *MKI67, PCNA, CCNB1, MCM2,* and *TOP2A*.
DOI: https://doi.org/10.7554/eLife.32838.014

• Transparent reporting form
DOI: https://doi.org/10.7554/eLife.32838.015

## Major datasets

The following previously published datasets were used:

| Author(s) | Year | Dataset title | Dataset URL | Database, license, and accessibility information |
|---|---|---|---|---|
| Schmidt M, Böhm D, von Törne C, Steiner E, Puhl A, Pilch H, Lehr H, Hengstler JG, Kölbl H, Gehrmann M | 2008 | The humoral immune system has a key prognostic impact in node-negative breast cancer | https://www.ncbi.nlm.nih.gov/geo/query/acc.cgi?acc=GSE11121 | Publicly available at the NCBI Gene Expression Omnibus (accession no: GSE11121) |
| Pawitan Y, Bjohle J, Amler L, Borg A, Egyhazi S, Hall P, Han X, Holmberg L, Huang F, Klaar S, Liu ET, Miller LD, Nordgren H, Ploner A, Sandelin K, Shaw PM, Smeds J, Skoog L, Wedren S, Bergh J | 2005 | Gene expression of breast cancer tissue in a large population-based cohort of Swedish patients | https://www.ncbi.nlm.nih.gov/geo/query/acc.cgi?acc=GSE1456 | Publicly available at the NCBI Gene Expression Omnibus (accession no: GSE1456) |
| Symmans WF, Sotiriou C, Andre F, Regitnig P, Daxenbichler G, Hatzis C | 2010 | Endocrine Sensitivity Index Validation Dataset | https://www.ncbi.nlm.nih.gov/geo/query/acc.cgi?acc=GSE17705 | Publicly available at the NCBI Gene Expression Omnibus (accession no: GSE17705) |
| Wang Y, Klijn JG, Zhang Y, Sieuwerts AM | 2005 | Breast cancer relapse free survival | https://www.ncbi.nlm.nih.gov/geo/query/acc.cgi?acc=GSE2034 | Publicly available at the NCBI Gene Expression Omnibus (accession no: GSE2034) |
| Buffa FM, Winchester L | 2011 | Gene expression profiling of early primary breast cancer to identify prognostic markers and associated pathways | https://www.ncbi.nlm.nih.gov/geo/query/acc.cgi?acc=GSE22219 | Publicly available at the NCBI Gene Expression Omnibus (accession no: GSE22219) |
| Greco D, Heikkilä P, Blomqvist C, Nevalinna H | 2011 | 183 breast tumors from the Helsinki Univerisity Central Hospital with survival information | https://www.ncbi.nlm.nih.gov/geo/query/acc.cgi?acc=GSE24450 | Publicly available at the NCBI Gene Expression Omnibus (accession no: GSE24450) |
| Symmans WF, Esserman L, Vidaurre T | 2011 | Discovery cohort for genomic predictor of response and survival following neoadjuvant taxane-anthracycline chemotherapy in breast cancer | https://www.ncbi.nlm.nih.gov/geo/query/acc.cgi?acc=GSE25055 | Publicly available at the NCBI Gene Expression Omnibus (accession no: GSE25055) |
| Symmans WF, Holmes F, Vidaurre T, Martin M, Souchon E Citation(s) | 2011 | Validation cohort for genomic predictor of response and survival following neoadjuvant taxane-anthracycline chemotherapy in breast cancer | https://www.ncbi.nlm.nih.gov/geo/query/acc.cgi?acc=GSE25065 | Publicly available at the NCBI Gene Expression Omnibus (accession no: GSE25065) |
| Nevins JR | 2005 | Breast Cancer Dataset | https://www.ncbi.nlm.nih.gov/geo/query/acc.cgi?acc=GSE3143 | Publicly available at the NCBI Gene Expression Omnibus (accession no: GSE3143) |
| Sabatier R, Finetti P, Adelaïde J, Guille A, Lane L, Birnbaum D, Chaffanet M, Bertucci F | 2011 | Down-regulation of ECRG4, a candidate tumor suppressor gene in human breast cancer | https://www.ncbi.nlm.nih.gov/geo/query/acc.cgi?acc=GSE31448 | Publicly available at the NCBI Gene Expression Omnibus (accession no: GSE31448) |

| | | | | |
|---|---|---|---|---|
| Miller LD, Smeds J, George J, Vega VB, Vergara L, Ploner A, Pawitan Y, Hall P, Klaar S, Liu ET, Bergh J | 2005 | An expression signature for p53 in breast cancer predicts mutation status, transcriptional effects, and patient survival | https://www.ncbi.nlm.nih.gov/geo/query/acc.cgi?acc=GSE3494 | Publicly available at the NCBI Gene Expression Omnibus (accession no: GSE3494) |
| Wang DY, Leong WL | 2014 | Internal validation cohort of breast cancers for development of ClinicoMolecular Triad Classification | https://www.ncbi.nlm.nih.gov/geo/query/acc.cgi?acc=GSE45725 | Publicly available at the NCBI Gene Expression Omnibus (accession no: GSE45725) |
| Ivshina AV, George J, Senko O, Mow B, Putti TC, Smeds J, Lindahl T, Pawitan Y, Hall P, Nordgren H, Wong JE, Liu ET, Bergh J, Kuznetsov VA, Miller LD | 2006 | Genetic Reclassification of Histologic Grade Delineates New Clinical Subtypes of Breast Cancer | https://www.ncbi.nlm.nih.gov/geo/query/acc.cgi?acc=GSE4922 | Publicly available at the NCBI Gene Expression Omnibus (accession no: GSE4922) |
| Loi S, Haibe-Kains B, Desmedt C, Lallemand F, Tutt AM, Gillet C, Ellis P, Harris A, Bergh J, Foekens JA, Klijn JG, Larsimont D, Buyse M, Bontempi G, Delorenzi M, Piccart MJ, Sotiriou C | 2007 | Definition of clinically distinct molecular subtypes in estrogen receptor positive breast carcinomas using genomic grade | https://www.ncbi.nlm.nih.gov/geo/query/acc.cgi?acc=GSE6532 | Publicly available at the NCBI Gene Expression Omnibus (accession no: GSE6532) |
| Desmedt C, Piette F, Loi S, Wang Y, Lallemand F, Haibe-Kains B, Viale G, Delorenzi M, Zhang Y, Saghatchian M, Bergh J, Lidereau R, Ellis P, Harris A, Klijn JG, Foekens JA, Cardoso F, Piccart MJ, Buyse M, Sotiriou C | 2007 | Strong Time Dependence of the 76-Gene Prognostic Signature | https://www.ncbi.nlm.nih.gov/geo/query/acc.cgi?acc=GSE7390 | Publicly available at the NCBI Gene Expression Omnibus (accession no: GSE7390) |
| Marthandan S, Klement K, Priebe S, Groth M, Platzer M, Hemmerich P, Diekmann S | 2016 | RNA-seq of human fibroblasts during replicative senescence | https://www.ncbi.nlm.nih.gov/geo/query/acc.cgi?acc=GSE63577 | Publicly available at the NCBI Gene Expression Omnibus (accession no: GSE63577) |

| | | | | |
|---|---|---|---|---|
| Uhlén M, Fagerberg L, Hallström BM, Lindskog C, Oksvold P, Mardinoglu A, Sivertsson Å, Kampf C, Sjöstedt E, Asplund A, Olsson I, Edlund K, Lundberg E, Navani S, Szigyarto CA, Odeberg J, Djureinovic D, Takanen JO, Hober S, Alm T, Edqvist PH, Berling H, Tegel H, Mulder J, Rockberg J, Nilsson P, Schwenk JM, Hamsten M, von Feilitzen K, Forsberg M, Persson L, Johansson F, Zwahlen M, von Heijne G, Nielsen J, Pontén F | 2015 | RNA-seq of coding RNA from tissue samples of 122 human individuals representing 32 different tissues | https://www.ebi.ac.uk/arrayexpress/experiments/E-MTAB-2836/ | Publicly available at the ArrayExpress (accession no: E-MTAB-2836) |
| Turcan S, Rohle D, Goenka A, Walsh LA | 2012 | IDH1 Mutation is a Master Regulator of Epigenomic Remodeling and is Sufficient to Establish the Glioma Hypermethylator Phenotype | https://www.ncbi.nlm.nih.gov/geo/query/acc.cgi?acc=GSE30339 | Publicly available at the NCBI Gene Expression Omnibus (accession no: GSE30339) |
| Abbas AR, Baldwin D, Ma Y, Ouyang W, Gurney A, Martin F, Fong S, van Lookeren Campagne M, Godowski P, Williams PM, Chan AC, Clark HF | 2010 | Expression profiles from a variety of resting and activated human immune cells | https://www.ncbi.nlm.nih.gov/geo/query/acc.cgi?acc=GSE22886 | Publicly available at the NCBI Gene Expression Omnibus (accession no: GSE22886) |

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
