## [Decision Letter]

Thank you for submitting your article "Combining CRISPR/Cas9 mutagenesis and a small-molecule inhibitor to probe the function of MELK in cancer" for consideration by *eLife*. Your article has been reviewed by two peer reviewers, and the evaluation has been overseen by a Reviewing Editor and Charles Sawyers as the Senior Editor. The following individuals involved in review of your submission have agreed to reveal their identity: Tony Hunter (Reviewer #1); David Stokoe (Reviewer #2).

The reviewers have discussed the reviews with one another and the Reviewing Editor has drafted this decision to help you prepare a revised submission.

Summary:

The authors have extended their analysis of the role of the MELK protein kinase in breast cancer cells. In their 2016 *eLife* paper, they re-examined the role of the MELK kinase, a mitotic kinase reported to be essential for basal breast carcinoma cell proliferation and be the target of OTS167, a small molecule MELK kinase inhibitor, in basal breast carcinoma (BBC). They found that CRISPR/Cas9-generated MELK knockout clones of two different TNBC breast cancer lines exhibited WT cell levels of proliferation, survival and anchorage independent growth and that MELK knockout cells remained sensitive to OTS167. They concluded that MELK is not required for TNBC cell proliferation and that OTS167 blocks growth in a MELK independent manner.

In this new study, they showed that stable overexpression of MELK in 3T3, Rat1-p53dd, and MCF10A cells did not promote colony growth or induce growth in soft agar. They also showed that colony formation and growth in soft agar of MELK-null clones of DLD1 (colorectal cancer), A375 (melanoma), Cal51 (breast cancer) and MDA-MB-231 (breast cancer) cells was similar to that of cells in which the Rosa26 locus had been targeted as a control; for Cal51 and MDA-MB-231 they showed that the MELK-null clones formed mammospheres as efficiently control cells. In addition, they found that the Cal51 and MDA-MB-231 MELK-null cells grew as well as Rosa26 control cells as flank xenografts in nude mice. Moreover, the MELK-null Cal51 and MDA-MB-231 cells were equally sensitive/resistant to H2O2, paclitaxel, idarubicin, bleomycin, oxaliplatin, and fluorouracil treatment, as their control counterparts. They also showed that acute inhibition of MELK in WT Cal51 and MDA-MB-231 cells or their MELK knockout derivatives with the highly-specific HTH-01-091MELK inhibitor caused no defect in proliferation. Finally, they re-analyzed publicly available data for a correlation between MELK expression and prognosis for breast cancer patients, and came to the conclusion that MELK expression is simply a marker for mitotic cells, and would therefore be expected to correlate with poor prognosis. These new data provide further evidence that the MELK kinase is not required for the tumorigenic properties of a variety of human cancer cell lines, including, at least for Cal51 cells, tumorigenesis.

Overall, this is a well-designed and executed study that confirms the previous data from this group and the Gray group showing that the multiple previous reports of MELK requirement in tumor cells were likely plagued by off-target effects of RNAi and small molecule reagents. This is therefore a useful addition to the small body of literature supporting this conclusion that can balance the >30 papers proposing a MELK requirement.

Essential revisions:

1) Although the Cal51 xenograft data are reasonable, the MDA-MB-231 xenograft data in Figure 2 are simply not convincing The Rosa 26 c1 cells did not grow at all (this is unexpected given that MDA-MB-231 cells generally grow very aggressively as xenograft tumors), whereas the MDA-MB-231 KO c1 cell tumors grew reasonably well. In contrast, both the MDA-MB-231 KO c2 and Rosa26 c2 cells grew poorly and with odd kinetics. The authors need to clearly acknowledge these imperfections in their own data and explain why the MDA-MB-231 cell results look so strange.

2) The Sanger sequencing of the PCR product from the knockout clones as shown is not a very informative presentation of the data. It would be more rigorous to pick ~10 clones of the PCR product after topo cloning and sequence these to get a more accurate representation of the gRNA-mediated alterations. Are the alterations shown homozygous, heterozygous, or hemizygous?

[Editors' note: further revisions were requested prior to acceptance, as described below.]

Thank you for resubmitting your work entitled "Combining CRISPR/Cas9 mutagenesis and a small-molecule inhibitor to probe the function of MELK in cancer" for further consideration at *eLife*. Your revised article has been favorably evaluated by Charles Sawyers (Senior Editor) and a Reviewing Editor.

The revised manuscript has been improved; however, the title, as written, implies that a new function of MELK has been revealed, and does not adequately describe the key conclusion. While the key conclusion is actually quite similar to that reflected in the title of your previous *eLife* article, it would be more appropriate to change the title to something along those lines.

---

## [Author Response]

Essential revisions:1) Although the Cal51 xenograft data are reasonable, the MDA-MB-231 xenograft data in Figure 2 are simply not convincing The Rosa 26 c1 cells did not grow at all (this is unexpected given that MDA-MB-231 cells generally grow very aggressively as xenograft tumors), whereas the MDA-MB-231 KO c1 cell tumors grew reasonably well. In contrast, both the MDA-MB-231 KO c2 and Rosa26 c2 cells grew poorly and with odd kinetics. The authors need to clearly acknowledge these imperfections in their own data and explain why the MDA-MB-231 cell results look so strange.

MDA-MB-231 is a highly-heterogeneous breast cancer cell line. Sub-populations within MDA-MB-231 exhibit significant differences in gene expression, proliferation, and colony-formation ability. For instance, Khan et al., 2017, derived single-cell clones from 12 MDA-MB-231 cells, and observed striking differences in their growth rates. Sub-populations within MDA-MB-231 are also known to exhibit varying levels of “stem cell-like” or “tumor-initiating” behavior (Fillmore et al., 2008). We hypothesized that the poor growth of our MDA-MB-231 clones as xenografts could simply reflect the fact that cells in the parental population display different abilities to form tumors, and, by chance, we isolated several clones with limited tumor initiating capacity. To test this, we performed additional xenografts using non-clonally-derived MDA-MB-231 populations (Figure 2—figure supplement 2). For these experiments, we transduced the MDA-MB-231 parental population with a Rosa26 gRNA or two different gRNAs targeting MELK, and then selected gRNA-expressing cells without single-cell cloning. We verified by western blotting that the transduced cells exhibited near-complete depletion of MELK (Figure 2—figure supplement 2). When these cell populations were injected into nude mice, they grew significantly faster than our MDA-MB-231 clones grew, and the animals had to be sacrificed within 26 days after injection. Additionally, the MELK-depleted populations grew at comparable or superior rates to the control populations. These results suggest that the poor growth of our clonal xenografts is due to differences in tumor-formation potential between single cells from MDA-MB-231, and further verify that MELK is dispensable for breast cancer growth.

2) The Sanger sequencing of the PCR product from the knockout clones as shown is not a very informative presentation of the data. It would be more rigorous to pick ~10 clones of the PCR product after topo cloning and sequence these to get a more accurate representation of the gRNA-mediated alterations. Are the alterations shown homozygous, heterozygous, or hemizygous?

We initially verified our A375 and DLD1 MELK-knockout clones by PCR-amplifying and sequencing the site targeted by the gRNA and by western blotting. As suggested by the reviewers, we further verified knockout status by using TOPO cloning to sequence individual MELK alleles from each clone (Figure 1—figure supplement 1). In total, 58 of 58 alleles that we sequenced had mutations at the locus targeted by the guide RNA, verifying on-target CRISPR cutting. DLD1 is a diploid cancer cell line with two copies of MELK, and our sequencing indicated that one MELK-KO clone has a homozygous 10bp deletion in MELK, while the other clone has different indels in each allele.

A375 is an aneuploid cell line with three copies of MELK. Two findings from this clone are worth noting. First, in one clone, we recovered seven different indel mutations in the MELK gene. We hypothesize that at the time of single-cell sorting, one allele in this cell had acquired a small deletion that did not fully abolish gRNA recognition. Then, during clonal expansion, this allele underwent additional mutagenesis to generate the multiple large indels that we recovered. Secondly, in another A375 clone, we identified one allele that had three different single-nucleotide substitutions that generated three independent missense mutations (E15V, T16I, I17L). For our guide design, we followed the strategy of Shi and Vakoc (Nature Biotech, 2015), and chose gRNA sequences that target conserved, functional protein domains. The guide present in this clone targets the MELK ATP binding domain, likely explaining why these missense mutations are sufficient to destabilize the protein despite the lack of an indel.

To provide additional evidence that all of our MELK-knockout clones lack detectable MELK protein, we performed western blotting using a second antibody that recognizes a distinct MELK epitope. We observed no protein expression in any MELK-KO clone using antibodies that recognize either the MELK N-terminus or the MELK C-terminus (Figure 2—figure supplement 1). In total, 100% of our topo-sequenced alleles harbor mutations in MELK, and western blotting with multiple antibodies failed to detect MELK, verifying that our MELK-KO clones lack wild-type MELK.

[Editors' note: further revisions were requested prior to acceptance, as described below.]

The revised manuscript has been improved; however, the title, as written, implies that a new function of MELK has been revealed, and does not adequately describe the key conclusion. While the key conclusion is actually quite similar to that reflected in the title of your previous eLife article, it would be more appropriate to change the title to something along those lines.

We have changed our manuscript's title to “MELK expression correlates with tumor mitotic activity but is not required for cancer growth”.